# The association between alcohol intake and incident atrial fibrillation in older adults: The ARIC cohort

**Louis Y. Li[1]\*, Linzi Li[1], Lin Yee Chen[2], Elsayed Z. Soliman[3], Alvaro Alonso[1]**

**1** Department of Epidemiology, Rollins School of Public Health, Emory University, Atlanta, GA, United States of America, **2** Lillehei Heart Institute and Department of Medicine (Cardiovascular Division), University of Minnesota Medical School, Minneapolis, MN, United States of America, **3** Department of Medicine, Epidemiological Cardiology Research Center, Section on Cardiovascular Medicine, Wake Forest School of Medicine, Winston-Salem, NC, United States of America

\* louis.li@emory.edu

**Data Availability Statement:** Data can be requested through the Biologic Specimen and Data Repository Information Coordinating Center (BioLINCC) website (https://biolincc.nhlbi.nih.gov/studies/aric/) after creating an account and

## Abstract

The association of alcohol intake with incident atrial fibrillation (AF) remains controversial, particularly among older adults. This study explores the association of alcohol intake with incident AF in older adults in the Atherosclerosis Risk in Communities (ARIC) cohort. Data were obtained from ARIC, a community-based cohort aimed to identify risk factors for cardiovascular disease. Alcohol intake was assessed through interviewer-administered questionnaires. Incident AF was ascertained between the 2011–2013 visit and 2019. Participants were classified as current, former, or never drinkers. Former drinkers were further categorized on weekly alcohol consumption: light, moderate, heavy. Covariates included demographic characteristics, prevalent cardiovascular disease, and other risk factors. The association between drinking characteristics and incident AF was analyzed using Cox proportional hazard models. There were 5,131 participants with mean (SD) age 75.2 (5.0) years, 41% male, 23% Black, and 739 (14%) cases of incident AF. Current and former drinkers had a similar risk of AF compared to never drinkers (HR 1.04, 95% CI 0.83–1.29; HR 1.16, 95% CI 0.93–1.45). Within former drinkers, heavy and moderate drinkers had a similar risk compared to light drinkers (HR 1.14, 95% CI 0.84–1.55; HR 1.15, 95% CI 0.75–1.78). AF risk did not differ per 5-year increase in years of abstinence (HR 1.00, 95% CI 0.96–1.03) or drinking (HR 1.07, 95% CI 0.96–1.19). We did not find consistent evidence supporting an increased risk of AF associated with alcohol intake in older adults, highlighting the need to further explore this relationship in older populations.

## Introduction

Atrial fibrillation (AF) is the most common cardiac arrhythmia, and its incidence and prevalence continue to increase worldwide [1]. Alcohol intake has been proposed as a potential risk factor in AF incidence in both acute drinking settings and in long-term consumption [2–4]. Alcohol consumption has direct effects on cellular, autonomic, and electrophysiological

registering with the site. The data dictionary is available on this website. More information about the ARIC study can be found at https://aric.cscc.unc.edu/aric9/. We confirm that these are third party data, and anyone would be able to access the data in the same manner as the authors. The authors did not have special access privileges that others would not have.

**Funding:** The Atherosclerosis Risk in Communities study has been funded in whole or in part with Federal funds from the National Heart, Lung, and Blood Institute, National Institutes of Health, Department of Health and Human Services, under Contract nos. (75N92022D00001, 75N92022D00002, 75N92022D00003, 75N92022D00004, 75N92022D00005). There was no additional funding received for this study.

**Competing interests:** The authors have declared that no competing interests exist.

functions involved in arrhythmogenicity [5]. Long-term alcohol consumption may directly increase AF risk through changes in the left atrial substrate including remodeling, dilation, and fibrosis as well as indirect effects through interactions with other AF risk factors such as hypertension and cardiomyopathy [6]. Additionally, alcohol intake may affect the autonomic nervous system, and it has been proposed that modulation of vagal and parasympathetic mechanisms may be involved in the pathogenesis of AF [7].

Previous studies have mostly shown positive associations between alcohol consumption and incident AF; however, more detailed characteristics of alcohol consumption, such as amount of alcohol consumed, years of drinking, and years of abstinence, have not been well-explored [8–17]. Older adults remain an understudied population in scientific literature, and few studies have directly addressed alcohol intake in older adults. Mukamal et al. [18] studied older adults in the Cardiovascular Health Study (CHS) and concluded that current moderate alcohol consumption was not associated with risk of incident AF but that former drinking was associated with higher risk of incident AF compared to never drinkers.

This study aims to further investigate the understudied population of older adults who are ≥65 years and are at increased risk of incident AF. Additionally, we hope to better characterize the relationship between more detailed characteristics of alcohol consumption among former drinkers including amount of alcohol consumed, years of abstinence, and years of drinking. We hypothesize that former drinkers and current drinkers will have a higher risk of incident AF than non-drinkers. We hypothesize that, among former drinkers, the years of abstinence will be associated with decreased risk of incident AF and the years of drinking and amount of alcohol consumed will be associated with increased risk of incident AF.

## Methods

### Study population

The ARIC study is a community-based cohort study that has been ongoing since 1987 to identify risk factors for atherosclerosis and cardiovascular disease. The study recruited individuals aged 45–64 years from 4 field centers: Washington County, MD; Forsyth County, NC; Jackson, MS; and selected suburbs of Minneapolis, MN. In total, 15,792 participants were enrolled at Visit 1 between 1987–1989 which included examinations for medical, social, and demographic data. Participants underwent periodic follow-up examination at designated intervals and were surveilled for cardiovascular outcomes. Visit 5 examinations took place between 2011–2013 and involved 6,538 participants, all of whom were ≥65 years, which served as the baseline for this study. Due to low sample sizes, two groups were excluded: (1) non-White and non-Black participants and (2) non-White participants from the Minneapolis and Washington County centers. Additional exclusion criteria included participants with missing information on alcohol consumption history at Visit 5, and prevalent AF. Prevalent AF was ascertained via ECGs performed at study visits and hospital discharge codes (ICD-9-CM: 427.3x; ICD-10-CM: I48.x) prior to Visit 5. Patients with missing covariates were also excluded, except for smoking status due to >10% of patients with missing smoking status. Statistical analysis was conducted for 5,131 participants that met study criteria (Fig 1).

The ARIC study was approved by the institutional review boards at University of Minnesota, Johns Hopkins University, Wake Forest University, University of North Carolina, University of Texas Health Sciences Center at Houston, and the University of Mississippi Medical Center. All participants provided written informed consent. Approval for the present analysis was obtained from Rollins School of Public Health at Emory University and the de-identified dataset was accessed on October 30, 2022.

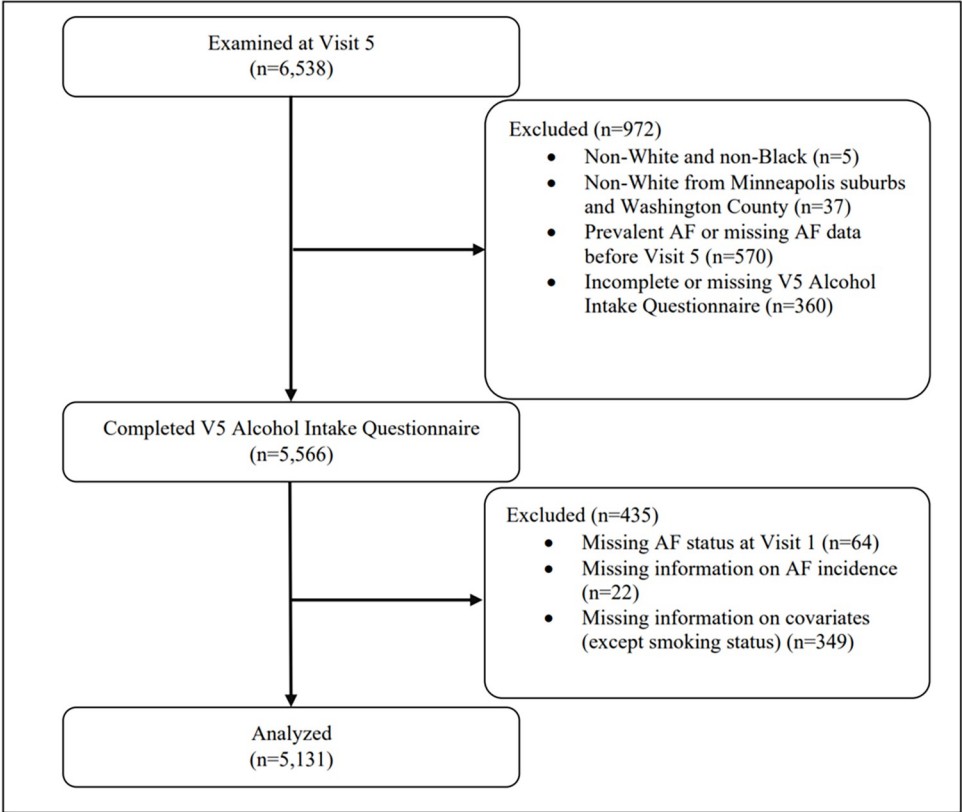

**Fig 1. Study population flowchart.**

## Assessment of incident atrial fibrillation

Incident AF was defined as any AF occurring after the participant's Visit 5 examination (2011–2013) and before December 31, 2019 for 3 centers (Washington County, Forsyth County, Minneapolis suburbs) and before December 31, 2017 for Jackson, MS. Incident AF was ascertained via hospital discharge codes (ICD-9-CM: 427.3x; ICD-10-CM: I48.x) not occurring in the context of open cardiac surgery and from death certificates with AF as an underlying or contributing cause of death (ICD-10: I48) [19].

## Assessment of alcohol intake

Alcohol intake was assessed from the ARIC Smoking and Alcohol Use Form at Visit 5. Participants were asked the following: (1) if they had ever consumed alcoholic beverages, (2) if they presently drink alcoholic beverages, (3) approximately how many years ago they stopped drinking, and (4) how much and what type of alcohol they consumed daily and weekly. Participants were stratified according to their drinker status: current, former, and never drinker.

Former drinkers were defined as having no drinks reported at Visit 5. Characteristics of former drinkers—amount of alcohol consumed weekly, years of abstinence, and years of drinking—were assessed at the most recent visit where drinking was reported. Amount of alcohol consumed weekly was calculated as one drink equal to 4oz wine (10.8g), 12oz beer (13.2g), or 1.5oz hard liquor (15.1g) [11, 20]. Weekly alcohol consumption was stratified into 3 categories based on definitions by the CDC's National Center for Health Statistics [21]:

- Light drinker: ≤3 drinks per week

- Moderate drinker: >3 drinks but ≤7 drinks per week for women; >3 drinks but ≤14 drinks per week for men

- Heavy drinker: >7 drinks per week for women; >14 drinks per week for men

## Assessment of additional covariates

Age, sex, race, education level, prevalent cardiovascular disease [coronary artery disease (CAD), heart failure (HF), and stroke], hypertension (HTN), HDL-C, LDL-C, use of antihypertensive medications, use of anticoagulants, diabetes, smoking status, and body mass index (BMI) were considered as potential confounders. Sex, race, and education level were obtained through self-report at Visit 1. Education level was stratified into 5 categories: high school or less, high school graduate, vocational school, college, and graduate/professional school.

Prevalent CAD, HF, stroke, hypertension, HDL-C, LDL-C, use of antihypertensive medications, use of anticoagulants, diabetes, smoking history, and BMI were assessed during Visit 5. Prevalent CAD was defined by myocardial infarction (MI) indicated on baseline ECG, self-reported MI, or cardiac procedure. Prevalent HF was determined by criteria from the ARIC heart failure research committee: any adjudicated HF event, any first position ICD-9 code of 428.x before 2005, any physician report of HF, two subsequent instances of self-reported HF or HF medication use, or an instance of self-reported HF or HF medication use with an elevated NT-proBNP value >125 pg/mL from Visit 4 or Visit 5. Prevalent stroke was defined by self-report of stroke or adjudicated incident stroke between baseline and Visit 5. Hypertension was defined as systolic blood pressure ≥140 mmHg, diastolic blood pressure ≥90 mmHg, or use of antihypertensive medications. HDL-C and LDL-C (mg/dL) were measured in blood using standard procedures. Use of antihypertensive medications and anticoagulants were obtained through self-report of use in the past 4 weeks. Diabetes was defined as fasting glucose ≥126 mg/dL, non-fasting glucose ≥200 mg/dL, current use of diabetes medication, or self-reported physician diagnosis of diabetes. Smoking status was stratified into 3 categories: current, former, and never smoker. BMI (kg/m$^2$) was calculated from height and weight measurements.

## Statistical analysis

The association between drinker status (current vs. former vs. never) and incident AF was analyzed using Kaplan-Meier and cumulative incidence curves and compared using log-rank tests with adjustments for potential confounders. The proportional hazards assumption was assessed using Kaplan-Meier curves and log-minus-log survival plots for each covariate. Within former drinkers, the association between weekly alcohol consumption (light vs. moderate vs. heavy) and incident AF was analyzed using Kaplan-Meier and cumulative incidence curves and compared using log-rank tests with adjustments for potential confounders. Years of drinking and years of abstinence were analyzed as continuous variables using Cox proportional hazard models. Additionally, for years of abstinence, participants were stratified into quartiles and 20-year intervals; for years of drinking, participants were stratified into quartiles and 10-year intervals. Statistical analysis was conducted in SAS software (Version 9.4; SAS Institute, Cary, NC, US).

## Results

### Demographic characteristics

There were 5,131 participants included with mean (SD) age of 75.2 (5.0) years, 41% male, and 23% Black (Table 1). Of the 5,131 participants, there were 2,544 (50%) current, 1,475 (29%)

**Table 1. Demographic characteristics of study population (n = 5,131).**

| Characteristic | Current Drinkers | Former Drinkers | Never Drinkers |
|---|---|---|---|
| | (n = 2,544) | (n = 1,475) | (n = 1,112) |
| Mean age, years (SD) | 74.9 (4.9) | 75.4 (5.1) | 75.8 (5.2) |
| Male, n (%) | 1202 (47) | 657 (45) | 239 (21) |
| Race, n (%) | | | |
| White | 2315 (91) | 942 (64) | 672 (60) |
| Black | 229 (9) | 533 (36) | 440 (40) |
| Education level, n (%) | | | |
| High school or less | 136 (5) | 312 (21) | 246 (22) |
| High school graduate | 764 (30) | 511 (35) | 424 (38) |
| Vocational school | 236 (9) | 118 (8) | 87 (8) |
| College | 986 (39) | 369 (25) | 225 (20) |
| Graduate/professional school | 422 (17) | 165 (11) | 130 (12) |
| Median drinks per week, n (IQR) | | | |
| Male | 4 (7) | 5 (30) | - |
| Female | 2 (5) | 0 (3) | - |
| Mean BMI, kg/m$^2$ (SD) | 27.9 (5.0) | 29.4 (5.9) | 29.3 (6.5) |
| Coronary artery disease, n (%) | 335 (13) | 222 (15) | 110 (10) |
| Heart failure, n (%) | 176 (7) | 218 (15) | 148 (13) |
| Stroke, n (%) | 57 (2.2) | 64 (4.3) | 36 (3.2) |
| Hypertension, n (%) | 1783 (70) | 1151 (78) | 862 (78) |
| Diabetes mellitus, n (%) | 638 (25) | 573 (39) | 393 (35) |
| HDL-C, mg/dL (SD) | 53.9 (14.5) | 49.7 (13.0) | 53.0 (13.1) |
| LDL-C, mg/dL (SD) | 105.4 (34.3) | 102.2 (34.1) | 108.9 (36.0) |
| Antihypertensive use, n (%) | 1559 (61) | 1058 (72) | 806 (72) |
| Anticoagulant use, n (%) | 73 (2.9) | 48 (3.3) | 37 (3.3) |
| Smoking status, n (%) | | | |
| Current smoker | 177 (7.0) | 98 (6.6) | 27 (2.4) |
| Former smoker | 1406 (55) | 809 (55) | 250 (22) |
| Never smoker | 821 (32) | 467 (32) | 744 (67) |
| Unknown/missing | 140 (5.5) | 101 (6.9) | 91 (8.2) |

former, and 1,112 (22%) never drinkers. Within current drinkers, there were 1,480 (58%), 747 (29%), and 298 (12%) light, moderate, and heavy drinkers. Within former drinkers, there were 882 (62%), 163 (11%), and 386 (27%) light, moderate, and heavy drinkers. Proportion of males was higher in current and former drinkers compared to never drinkers (47%, 45%, 21%, respectively). Lastly, never drinkers were more frequently likely to be never smokers compared to current and former drinkers (67%, 32%, 32%, respectively).

## AF incidence

In total, 739 (14%) participants had incident AF: 377 (15%) current drinkers, 226 (15%) former drinkers, and 136 (12%) never drinkers. With an alpha level of 0.05, there was a statistically significant difference in the risk of incident AF between categories due to the difference in risk between former and never drinkers (Fig 2). The difference in risk between former vs. never drinkers was statistically significant (p = 0.006). The differences between current vs. former and current vs. never drinkers were not statistically significant (p = 0.067 and 0.152, respectively). Participants were followed for a median time of 7.0 years. The overall incidence of AF

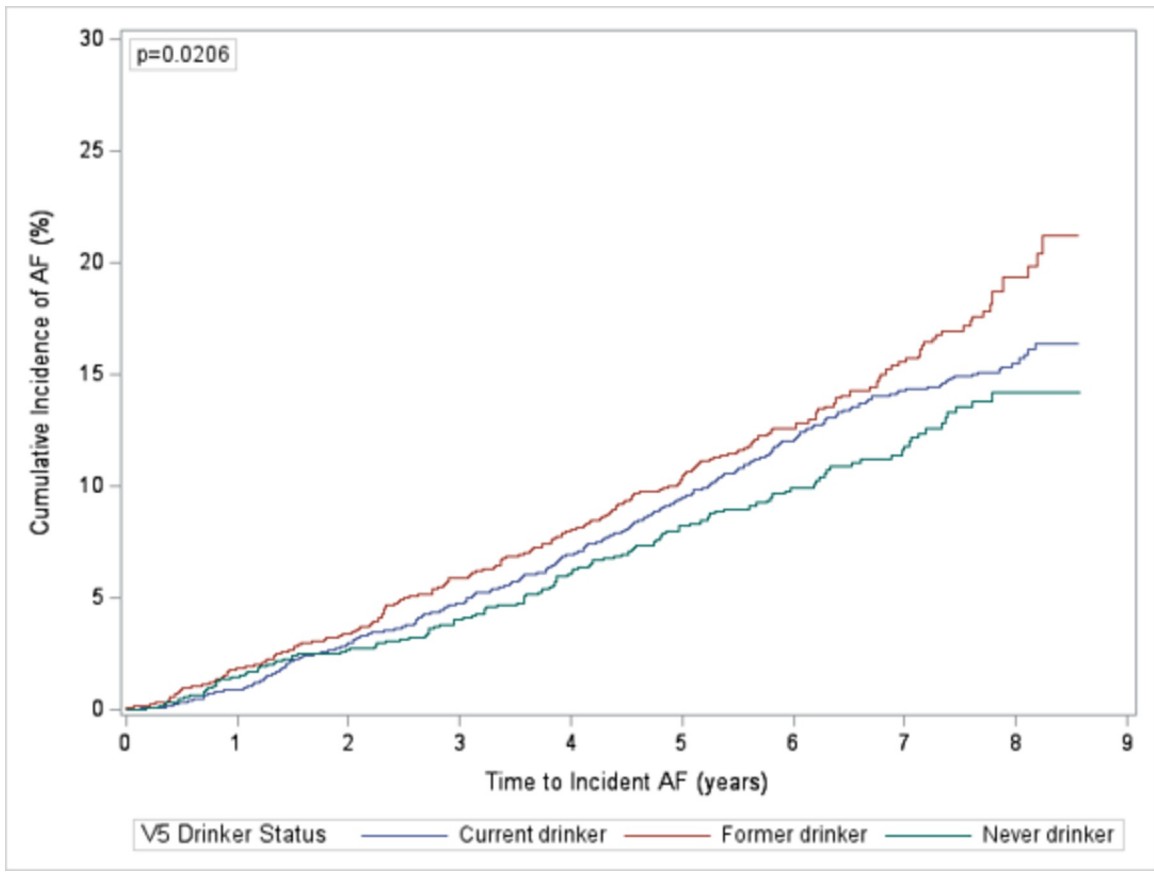

**Fig 2. Cumulative incidence of atrial fibrillation by drinker status.** Cumulative incidence of atrial fibrillation by drinker status comparing current, former, and never drinkers.

was 23.2 cases per 1,000 person-years (PY): current drinkers (23.0 cases per 1,000 PYs), former drinkers (25.8 cases per 1,000 PYs), never drinkers (20.4 cases per 1,000 PYs).

## Drinker status

In unadjusted analyses, current drinkers had a higher risk of incident AF compared to never drinkers; however, the result was not statistically significant (HR 1.12, 95% CI 0.92–1.37). There was not a statistically significant difference in the risk of incident AF between categories of current drinkers (Fig 3). Former drinkers had a higher risk of incident AF compared to never drinkers (HR 1.27, 95% CI 1.02–1.57). After adjusting for covariates, the risk was attenuated in both current and former drinkers, and the risk ratios were not statistically significant (HR 1.04, 95% CI 0.83–1.29; HR 1.16, 95% CI 0.93–1.45, respectively) (Table 2).

## Former drinkers

Within former drinkers, 1,431 out of 1,475 participants had data available on the number of drinks consumed weekly and were stratified into 3 categories: light, moderate, and heavy drinkers. There were 882 (62%) light drinkers, 163 (11%) moderate drinkers, and 386 (27%) heavy drinkers. There were 221 (15%) cases of incident AF: 128 (15%) occurring in light drinkers, 28 (17%) occurring in moderate drinkers, and 65 (17%) occurring in heavy drinkers. On average, light, moderate, and heavy drinkers consumed 0, 7, and 45 drinks per week for men

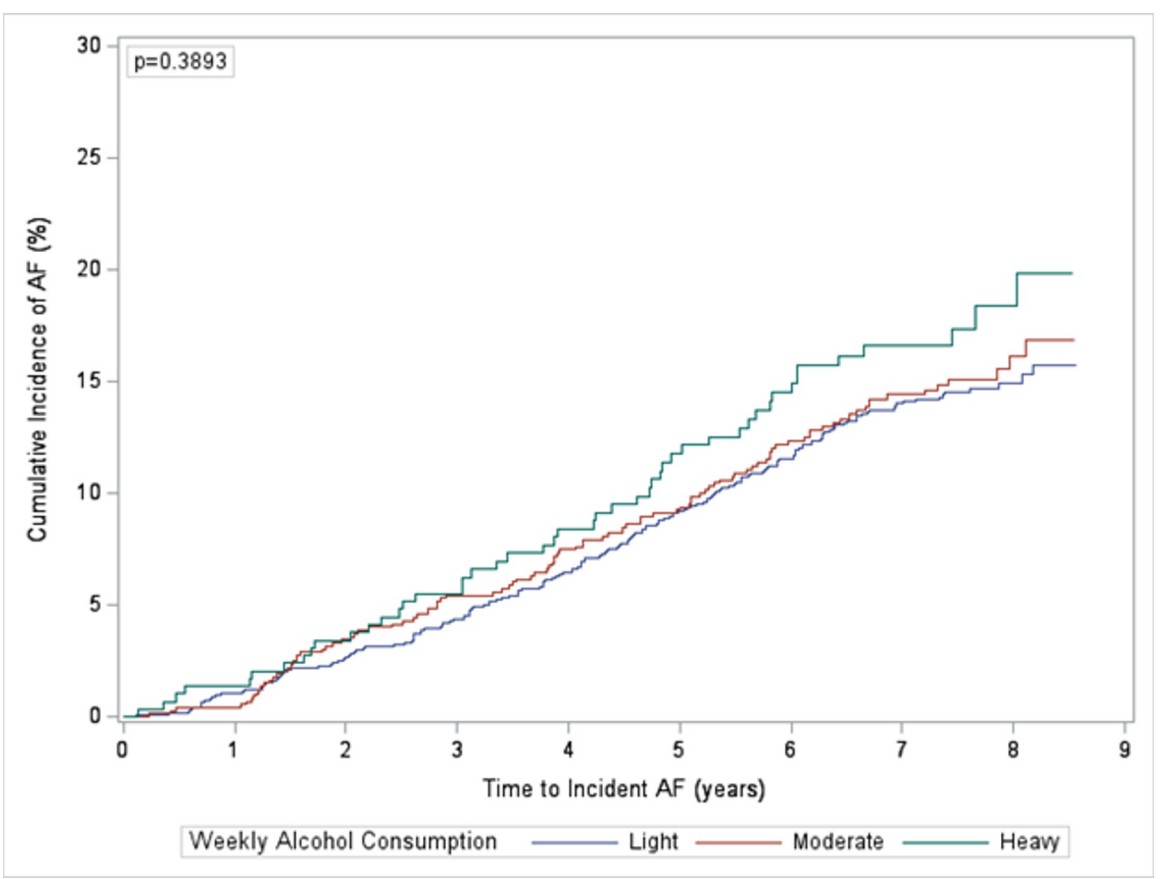

**Fig 3. Cumulative incidence of atrial fibrillation by weekly alcohol consumption in current drinkers.** Cumulative incidence of atrial fibrillation by weekly alcohol consumption in current drinkers comparing light, moderate, and heavy drinkers.

and 0, 5, and 26 drinks per week for women. There was not a statistically significant difference in the risk of incident AF between categories of former drinkers (Fig 4). In unadjusted analyses, heavy and moderate drinkers had a higher risk of incident AF compared to light drinkers (HR 1.25, 95% CI 0.93–1.69; HR 1.22, 95% CI 0.81–1.83). In adjusted analyses, the comparisons were attenuated for both heavy (HR 1.14, 95% CI 0.84–1.55) and moderate (HR 1.15, 95% CI 0.75–1.78) drinkers (Table 3).

Incident AF in former drinkers was also analyzed on the number of years of abstinence and number of years of drinking for 1,393 and 676 participants, respectively, who had available data. The median (IQR) years of abstinence was 20 (30) years, the range was 0–74 years, and there were 214 (15%) cases of incident AF. The median (IQR) years of drinking was 10 (15)

**Table 2. Risk of incident atrial fibrillation by drinker status (n = 5,131).**

|  | AF Cases | Incidence Rate (per 1,000 PYs) | Unadjusted Hazard Ratio | 95% Confidence Interval | Adjusted Hazard Ratio[a] | 95% Confidence Interval |
|---|---|---|---|---|---|---|
| Never | 136 (12%) | 20.4 | 1 (Ref.) | Ref. | 1 (Ref.) | Ref. |
| Current | 377 (15%) | 23.0 | 1.12 | 0.92–1.37 | 1.04 | 0.83–1.29 |
| Former | 226 (15%) | 25.8 | 1.27 | 1.02–1.57 | 1.16 | 0.93–1.45 |

[a]Adjusted for age, sex, race, education level, prevalent cardiovascular disease [coronary artery disease (CAD), heart failure (HF), and stroke], hypertension (HTN), HDL-C, LDL-C, use of antihypertensive medications, use of anticoagulants, diabetes, smoking status, and body mass index (BMI).

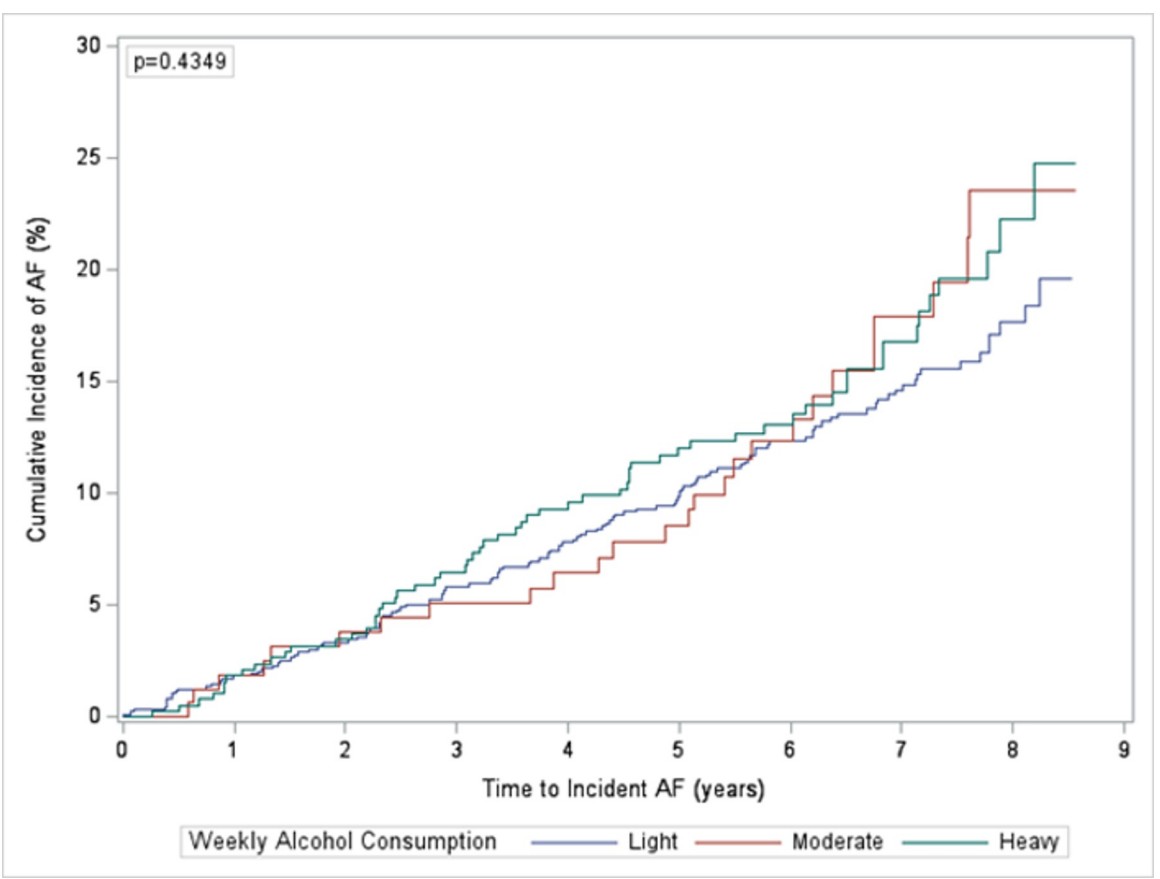

**Fig 4. Cumulative incidence of atrial fibrillation by weekly alcohol consumption in former drinkers.** Cumulative incidence of atrial fibrillation by weekly alcohol consumption in former drinkers comparing light, moderate, and heavy drinkers.

years, the range was 0–43 years, and there were 102 (15%) cases of incident AF. There was no difference in the risk of incident AF per 5 years of abstinence in unadjusted (HR 1.00, 95% CI 0.96–1.03) or adjusted (HR 0.98, 95% CI 0.94–1.01) analyses. There was a small increase in the risk of incident AF per 5 years of drinking in unadjusted (HR 1.10, 95% CI 0.99–1.22) that was attenuated in adjusted analysis (HR 1.07, 95% CI 0.96–1.19) (Table 4). In unadjusted and adjusted analyses by quartile and by 10-year and 20-year intervals did not yield statistically significant results for years of abstinence or years of drinking (S1–S4 Tables).

**Table 3. Risk of incident atrial fibrillation by weekly alcohol consumption in former drinkers (n = 1,431).**

|  | AF Cases | Incidence Rate (per 1,000 PYs) | Unadjusted Hazard Ratio | 95% Confidence Interval | Adjusted Hazard Ratio[a] | 95% Confidence Interval |
|---|---|---|---|---|---|---|
| **Light** | 128 (15%) | 23.9 | 1 (Ref.) | Ref. | 1 (Ref.) | Ref. |
| **Moderate** | 28 (17%) | 28.0 | 1.22 | 0.81–1.83 | 1.15 | 0.75–1.78 |
| **Heavy** | 65 (17%) | 29.4 | 1.25 | 0.93–1.69 | 1.14 | 0.84–1.55 |

[a]Adjusted for age, sex, race, education level, prevalent cardiovascular disease [coronary artery disease (CAD), heart failure (HF), and stroke], hypertension (HTN), HDL-C, LDL-C, use of antihypertensive medications, use of anticoagulants, diabetes, smoking status, and body mass index (BMI).

**Table 4. Risk of incident atrial fibrillation by years of abstinence and drinking in former drinkers (n = 1,393; n = 676).**

|  | Unadjusted Hazard Ratio | 95% Confidence Interval | Adjusted Hazard Ratio[a] | 95% Confidence Interval |
|---|---|---|---|---|
| **Years of abstinence** | 1.00[b] | 0.96–1.03 | 0.98[b] | 0.94–1.01 |
| **Years of drinking** | 1.10[b] | 0.99–1.22 | 1.07[b] | 0.96–1.19 |

[a]Adjusted for age, sex, race, education level, prevalent cardiovascular disease [coronary artery disease (CAD), heart failure (HF), and stroke], hypertension (HTN), HDL-C, LDL-C, use of antihypertensive medications, use of anticoagulants, diabetes, smoking status, and body mass index (BMI).
[b]Hazard ratio per 5 years.

## Discussion

In a community-based cohort study exploring the association between past alcohol intake and incident AF in older adults, the overall incidence of AF was 23.2 cases per 1,000 person-years, which is consistent with previous literature estimating the incidence in older adults between 14.9–28.3 per 1,000 person-years [22–24]. Current drinkers had a 4% higher risk of incident AF compared to never drinkers and former drinkers had a 12% higher risk of incident AF compared to never drinkers in adjusted analysis; however, these results were not statistically significant. While we hypothesized current drinkers to have a higher risk than former drinkers, a possible explanation could involve participants' reasons for quitting drinking such as underlying health conditions, including other AF risk factors, that may explain the lack of significant differences between groups. For example, 13% and 15% of never and former drinkers had prevalent HF, while only 7% of current drinkers had prevalent HF. People with a greater number of comorbidities or risk factors may choose to never drink or quit drinking, which may explain the lack of statistical significantly different results.

Several studies have identified a positive association between alcohol consumption and incident AF; however, most studies focus on middle-aged adults rather than on older adults [8–17]. Few studies have focused on this relationship in older adults; however, the results have been inconclusive. In the Cardiovascular Health Study (CHS), Mukamal et al. [18] demonstrated that current moderate alcohol consumption was not associated with the risk of incident AF, but former drinkers were at higher risk than never drinkers. In an older Chinese population, Ye et al. [25] concluded that current drinkers were at higher risk than never drinkers in women only and not men. The results from this study show weak associations between drinker status and incident AF and do not support a strong effect of alcohol intake on AF risk in older adults. The relationship in older adults is likely multifaceted and complex, possibly depending on alcohol drinking patterns, prevalence of other risk factors, differences in the underlying risk of AF, and methodological features including methods of assessing alcohol consumption (such as greater recall bias in older populations) and incident AF, thus requiring further investigation into the direct mechanisms of alcohol consumption on AF pathogenesis.

Within former drinkers, moderate and heavy drinkers had a higher risk of incident AF compared to light drinkers, and heavy drinkers had a similar risk to moderate drinkers; however, these results were not statistically significant. Similar to the comparisons between current vs. former vs. never drinkers, advanced age and underlying comorbidities may confound the true association between alcohol intake and incident AF as other risk factors may be more impactful. Additionally, older adults may drink less as they age, further confounding the association. Studies addressing more detailed amount of alcohol consumed suggest an association between increasing alcohol consumption and incident AF. In the Framingham Heart Study (FHS) with a cohort aged 28–62 years, there was weak association between long-term moderate alcohol consumption and the risk of incident AF, but there was a significant association between heavy consumption and incident AF [3]. A separate study conducted in the FHS on a

middle-aged population determined that every 10 g per day of additional alcohol consumed was associated with a 5% higher risk of incident AF [6]. A previous analysis on the ARIC cohort addressed baseline alcohol intake and incident AF in a younger population aged 45–64 years and demonstrated that greater alcohol consumption was associated with an increased risk of incident AF in persons aged 45–64 years [11]. Previous literature suggests that heavy alcohol consumption may increase the risk of incident AF in younger populations; however, the findings from this study are not conclusive in supporting this hypothesis in older adults.

In this study focusing on older adults, years of abstinence were not associated with an increased risk of incident AF. Each 5-year increase in years of drinking was associated with a 7% increased risk of incident AF; however, this association was not statistically significant. The previous ARIC analysis by Dixit et al. [11] demonstrated that a longer duration of alcohol abstinence among former drinkers was associated with a lower risk of incident AF and greater alcohol consumption was associated with a higher risk of incident AF in persons aged 45–64 years. The variable conclusions between this study and Dixit et al. [11] may suggest a fundamental difference between the study populations because of age. In the younger population, alcohol consumption may be a greater risk factor for incident AF; however, in older adults, alcohol consumption may not play as large of a role in the development of AF and may be masked by other, more significant risk factors associated with aging. To better characterize these differences, future investigations may compare younger and older populations as well as identifying the specific pathophysiologic changes that occur in the context of alcohol consumption.

Strengths of this study include extensive follow-up time and the quality of assessments of alcohol intake and additional covariates. Furthermore, to our knowledge, this study is only the second U.S.-based investigation in a population of older adults on the association between alcohol consumption and incident AF. There were several limitations in this study. First, assessments of alcohol intake were reliant on self-report data through questionnaires. Self-report of alcohol intake may introduce recall bias but has been previously identified as a reliable and valid approach, and careful attention was given to accurately represent the data [26, 27]. Additionally, the CDC guidelines on drinking categorizations may not be as applicable to older populations. Next, incident AF was ascertained from ICD codes and death certificates without ECG, Holter monitoring, or echocardiographic data. While this may affect our results by missing cases, such as in paroxysmal episodes where AF may be missed entirely, previous studies have validated the use of hospital discharge codes and death certificates for AF ascertainment [19, 28]. Data on the severity and type of AF, such as paroxysmal, persistent, or permanent were not captured, and these characteristics may be important distinctions in the effects of alcohol consumption on AF incidence. Data was not collected on participants' reasons for quitting drinking as health concerns may potentially explain the increased risk of incident AF among former drinkers. Last, while efforts were made to adjust for measured confounders, residual confounding may lead to bias in the reported results.

Our findings suggest that the association between alcohol intake and incident AF may be less conclusive and more multifaceted in older adults than in the general population. In this study of older adults, other risk factors may be more powerful predictors of AF that led to weaker associations between alcohol intake and AF. Future investigations in older populations should compare findings to younger populations, identify other potential risk factors that occur during the aging process that may be more involved in AF pathogenesis and may mask or supersede the effects of alcohol intake, and address concerns regarding the severity and type of AF that patients may experience. As the global burden of AF continues to rise, better understandings of AF are a necessary and worthwhile endeavor, especially in older populations who are at greatest risk.

## Supporting information

**S1 Table. Risk of incident atrial fibrillation by quartiles of years of abstinence in former drinkers (n = 1,393).** [a]Adjusted for age, sex, race, education level, prevalent cardiovascular disease [coronary artery disease (CAD), heart failure (HF), and stroke], hypertension (HTN), HDL-C, LDL-C, use of antihypertensive medications, use of anticoagulants, diabetes, smoking status, and body mass index (BMI).
(DOCX)

**S2 Table. Risk of incident atrial fibrillation by 20-year intervals of years in former drinkers (n = 1,393).** [a]Adjusted for age, sex, race, education level, prevalent cardiovascular disease [coronary artery disease (CAD), heart failure (HF), and stroke], hypertension (HTN), HDL-C, LDL-C, use of antihypertensive medications, use of anticoagulants, diabetes, smoking status, and body mass index (BMI).
(DOCX)

**S3 Table. Risk of incident atrial fibrillation by quartiles of years of drinking in former drinkers (n = 676).** [a]Adjusted for age, sex, race, education level, prevalent cardiovascular disease [coronary artery disease (CAD), heart failure (HF), and stroke], hypertension (HTN), HDL-C, LDL-C, use of antihypertensive medications, use of anticoagulants, diabetes, smoking status, and body mass index (BMI).
(DOCX)

**S4 Table. Risk of incident atrial fibrillation by 10-year intervals for years of drinking in former drinkers (n = 676).** [a]Adjusted for age, sex, race, education level, prevalent cardiovascular disease [coronary artery disease (CAD), heart failure (HF), and stroke], hypertension (HTN), HDL-C, LDL-C, use of antihypertensive medications, use of anticoagulants, diabetes, smoking status, and body mass index (BMI). [b]The category of 31–43 years includes 1 participant who had >40 years of drinking and was not separated into an additional category due to low sample size. All other participants had between 31–40 years of drinking.
(DOCX)

## Acknowledgments

The authors thank the staff and participants of the ARIC study for their important contributions.

## Author Contributions

**Conceptualization:** Louis Y. Li, Linzi Li, Alvaro Alonso.

**Data curation:** Louis Y. Li.

**Formal analysis:** Louis Y. Li.

**Investigation:** Louis Y. Li, Alvaro Alonso.

**Methodology:** Louis Y. Li, Linzi Li, Alvaro Alonso.

**Project administration:** Alvaro Alonso.

**Resources:** Louis Y. Li.

**Software:** Louis Y. Li.

**Supervision:** Alvaro Alonso.

**Visualization:** Louis Y. Li.

**Writing – original draft:** Louis Y. Li.

**Writing – review & editing:** Louis Y. Li, Linzi Li, Lin Yee Chen, Elsayed Z. Soliman, Alvaro Alonso.

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
