## [Decision Letter · Decision Letter 0]

29 Aug 2024

PONE-D-24-19326The Association between Alcohol Intake and Incident Atrial Fibrillation in Older Adults: The ARIC CohortPLOS ONE

Dear Dr. Louis Y Li,

Thank you for submitting your manuscript to PLOS ONE. After careful consideration, we feel that it has merit but does not fully meet PLOS ONE’s publication criteria as it currently stands. Therefore, we invite you to submit a revised version of the manuscript that addresses the points raised during the review process.

Please submit your revised manuscript by Oct 13 2024 11:59PM. If you will need more time than this to complete your revisions, please reply to this message or contact the journal office at plosone@plos.org. Please include the following items when submitting your revised manuscript:A rebuttal letter that responds to each point raised by the academic editor and reviewer(s). You should upload this letter as a separate file labeled 'Response to Reviewers'.A marked-up copy of your manuscript that highlights changes made to the original version. You should upload this as a separate file labeled 'Revised Manuscript with Track Changes'.An unmarked version of your revised paper without tracked changes. You should upload this as a separate file labeled 'Manuscript'.

We look forward to receiving your revised manuscript.

Kind regards,

Ricardas Radisauskas

Academic Editor

PLOS ONE

Journal Requirements:

"The Atherosclerosis Risk in Communities study has been funded in whole or in part with Federal funds from the National Heart, Lung, and Blood Institute, National Institutes of Health, Department of Health and Human Services, under Contract nos. (75N92022D00001, 75N92022D00002, 75N92022D00003, 75N92022D00004, 75N92022D00005)."

3. In the online submission form, you indicated that [Data are available through request from the ARIC Data Coordinating Center (contact via ARIChelp@unc.edu) for researchers who meet the criteria for access to confidential data. The data underlying the results presented in the study are available from the ARIC Data Coordinating Center at https://aric.cscc.unc.edu/aric9/researchers/Obtain_Submit_Data.

ARIC data can also be accessed via BioLINCC without the need for the ARIC Study approval at https://biolincc.nhlbi.nih.gov/. There may be some differences in the data available from BioLINCC, such as the removal of extreme values and the omission of restricted data.]. 

Reviewers' comments:

Reviewer's Responses to Questions

**Comments to the Author**

1. Is the manuscript technically sound, and do the data support the conclusions?

Reviewer #1: Yes

Reviewer #2: Yes

Reviewer #3: No

2. Has the statistical analysis been performed appropriately and rigorously? 

Reviewer #1: Yes

Reviewer #2: Yes

Reviewer #3: No

3. Have the authors made all data underlying the findings in their manuscript fully available?

Reviewer #1: Yes

Reviewer #2: No

Reviewer #3: No

4. Is the manuscript presented in an intelligible fashion and written in standard English?

Reviewer #1: Yes

Reviewer #2: Yes

Reviewer #3: No

5. Review Comments to the Author

Reviewer #1: This study assessed the association between alcohol intake and incident atrial fibrillation (AF) among older participants of the ARIC study (persons age >65 years).

1. The study has limitations in determination of the alcohol intake status (self-reported) and quantity of alcohol consumed (potential recall bias). This was appropriately acknowledged in the limitations.

2. There may also be problems in AF determination. The methodology of determining AF by ICD codes may miss paroxysmal episodes. I would be more explicit in acknowledging this limitation in the manuscript

3. In the Discussion section authors have tried to explain alcohol’s negative association with AF among older individuals vs. positive association among younger individuals. One possibility could be that in the older individuals other, more powerful predictors of AF drawn weaker predictors such as alcohol.

4. Why does Figure 2 have 4 panels when panel A is providing the whole picture? Furthermore, changing the color scheme in each panel is confusing. I would only present panel A

5. Line 356, “that” repeated twice

Reviewer #2: I have no particular concern about the manuscript. The argument is very interesting, the data are abundant and derive by a well-known study, English language is plain and clear, strength and limitations are well exposed.

Reviewer #3: -No echocardiographic data to diagnose the AF etiology and presence of structural heart disease

-Thrombotic risk as CHADS VASC score and presentation with systemic embolization as CVA is not included.

-No documented evidence of presence of AF as ECG or holter monitoring.

6. PLOS authors have the option to publish the peer review history of their article (what does this mean?). If published, this will include your full peer review and any attached files.

Reviewer #1: No

Reviewer #2: No

Reviewer #3: No

---

## [Author Response · Author response to Decision Letter 0]

13 Sep 2024

Reviewer #1

1. The study has limitations in determination of the alcohol intake status (self-reported) and quantity of alcohol consumed (potential recall bias). This was appropriately acknowledged in the limitations.

Acknowledged and no response required.

2. There may also be problems in AF determination. The methodology of determining AF by ICD codes may miss paroxysmal episodes. I would be more explicit in acknowledging this limitation in the manuscript

Included language to explicitly acknowledge limitations of AF acquisition: Next, due to incident AF being ascertained from ECGs and ICD codes and without echocardiographic data, the severity and type of AF, such as paroxysmal, persistent, or permanent were not captured. In cases such as paroxysmal episodes, incident AF may be missed entirely.

3. In the Discussion section authors have tried to explain alcohol’s negative association with AF among older individuals vs. positive association among younger individuals. One possibility could be that in the older individuals other, more powerful predictors of AF drawn weaker predictors such as alcohol.

Included language to clarify this statement: In this study of older adults, other risk factors may be more powerful predictors of AF that led to weaker associations between alcohol intake and AF.

4. Why does Figure 2 have 4 panels when panel A is providing the whole picture? Furthermore, changing the color scheme in each panel is confusing. I would only present panel A

The panels were initially provided to show the statistical significance of results between comparisons. Panel A showed that there was statistically significant difference when all 3 groups were compared together whereas Panels B-D show that the difference between the comparisons: current vs. former, current vs. never, and former vs. never. These additional panels show that there was not statistically significant difference between current vs. former and current vs. never drinkers, and that the main contributing reason for the statistically significant difference in Panel A is a result of the differences between former vs. never drinkers (Panel D, p<0.05). 

Changed Figure 2 to present only Panel A with an updated caption and additional explanation in the text as follows: With an alpha level of 0.05, there was a statistically significant difference in the risk of incident AF between categories due to the difference in risk between former and never drinkers (Fig 2). The difference in risk between former vs. never drinkers was statistically significant (p=0.006). The differences between current vs. former and current vs. never drinkers were not statistically significant (p=0.067 and 0.152, respectively).

5. Line 356, “that” repeated twice

Removed “that”.

Reviewer #2

I have no particular concern about the manuscript. The argument is very interesting, the data are abundant and derive by a well-known study, English language is plain and clear, strength and limitations are well exposed.

Acknowledged and no response required.

Reviewer #3

1. No echocardiographic data to diagnose the AF etiology and presence of structural heart disease

Even though echocardiographic data was collected in a subset of ARIC participants at visit 5 (baseline for this analysis), we do not have echocardiographic data at the time of AF diagnosis. We understand this lack of data to characterize AF etiology and determine the presence of structural heart disease is a limitation and we now note it in the Discussion as a limitation: Next, incident AF was ascertained from ICD codes and death certificates without ECG, Holter monitoring, or echocardiographic data. While this may affect our results by missing cases, such as in paroxysmal episodes where AF may be missed entirely, previous studies have validated the use of hospital discharge codes and death certificates for AF ascertainment.

2. Thrombotic risk as CHADS VASC score and presentation with systemic embolization as CVA is not included.

The primary focus of our analysis is on the link between alcohol consumption and development of new onset AF, rather than on the risk of thromboembolic events in people with AF. Thus, inclusion of the CHA2DS2-Vasc score and consideration of thromboembolic outcomes is not directly relevant to our primary study question.

3. No documented evidence of presence of AF as ECG or holter monitoring.

Prevalent AF was ascertained via ECGs performed at study visits and hospital discharge codes prior to baseline for this analysis, as Holter monitoring was not conducted as part of the ARIC examinations. Similarly, incident AF relied on hospital discharge codes and death certificates. This is not the gold standard for AF diagnosis, which would require evidence of the arrhythmia in an ECG recording. However, validation studies, including our work in the ARIC cohort (Alonso et al, 2009, PMID: 19540400; Jensen et al, 2012, PMID: 22262600), have demonstrated the excellent validity of this approach to identify AF in large epidemiologic studies in which Holter monitoring and repeated ECGs is not logistically feasible. We mention lack of ECG confirmation of all AF cases and Holter monitoring in the study as limitations that could affect our results: Next, incident AF was ascertained from ICD codes and death certificates without ECG, Holter monitoring, or echocardiographic data. While this may affect our results by missing cases, such as in paroxysmal episodes where AF may be missed entirely, previous studies have validated the use of hospital discharge codes and death certificates for AF ascertainment.19,28 Data on the severity and type of AF, such as paroxysmal, persistent, or permanent were not captured, and these characteristics may be important distinctions in the effects of alcohol consumption on AF incidence.

---

## [Editor Report · Decision Letter 1]

7 Nov 2024

The Association between Alcohol Intake and Incident Atrial Fibrillation in Older Adults: The ARIC Cohort

PONE-D-24-19326R1

Dear Dr. Louis Y Li,

We’re pleased to inform you that your manuscript has been judged scientifically suitable for publication and will be formally accepted for publication once it meets all outstanding technical requirements.

Kind regards,

Ricardas Radisauskas

Academic Editor

PLOS ONE
---

## [Editor Report · Acceptance letter]

11 Nov 2024

PONE-D-24-19326R1 

PLOS ONE

Dear Dr. Li, 

I'm pleased to inform you that your manuscript has been deemed suitable for publication in PLOS ONE. Congratulations! Your manuscript is now being handed over to our production team.

Kind regards, 

on behalf of

Professor Ricardas Radisauskas 

Academic Editor

PLOS ONE